# Small-Scale Foreign Object Debris Detection Using Deep Learning and Dual Light Modes

Yiming Mo [1], Lei Wang [1,*], Wenqing Hong [2], Congzhen Chu [3], Peigen Li [1] and Haiting Xia [1,*]

[1] Faculty of Civil Aviation and Aeronautics, Kunming University of Science and Technology, Kunming 650500, China; mym_1003@163.com (Y.M.); 20223110001@stu.kust.edu.cn (P.L.)
[2] Kunming Institute of Physics, Kunming 650223, China; hongwenqing@aliyun.com
[3] Yunnan Airport Group Co., Ltd., Kunming 650500, China; lehmannc@126.com
[*] Correspondence: wang_lei@kust.edu.cn (L.W.); htxia2006@163.com (H.X.)

**Abstract:** The intrusion of foreign objects on airport runways during aircraft takeoff and landing poses a significant safety threat to air transportation. Small-scale Foreign Object Debris (FOD) cannot be ruled out on time by traditional manual inspection, and there is also a potential risk of secondary foreign body intrusion. A deep-learning-based intelligent detection method is proposed to solve the problem of low accuracy and low efficiency of small-scale FOD detection. Firstly, a dual light camera system is utilized for the collection of FOD data. It generates a dual light FOD dataset containing both infrared and visible light images. Subsequently, a multi-attention mechanism and a bidirectional feature pyramid are integrated into the baseline network YOLOv5. This integration prioritizes the extraction of foreign object features and boosts the network's ability to distinguish FOD from complex backgrounds. Additionally, it enhances the fusion of higher-level features to improve the representation of multi-scale objects. To ensure fast and accurate localization and recognition of targets, the Complete-IoU (CIoU) loss function is used to optimize the bounding boxes' positions. The experimental results indicate that the proposed model achieves a detection speed of 36.3 frame/s, satisfying real-time detection requirements. The model also attains an average accuracy of 91.1%, which is 7.4% higher than the baseline network. Consequently, this paper verifies the effectiveness and practical utility of our algorithm for the detection of small-scale FOD targets.

**Keywords:** foreign object debris; small target detection; dual light; multi-attention; deep learning

## 1. Introduction

The runway is an essential infrastructure for aircraft takeoff and landing, affecting the airport's operational safety and support capacity. Foreign Object Debris (FOD) [1] poses a direct threat to aircraft safety and is a major safety hazard in air transportation. Therefore, real-time monitoring of airport runways, as well as real-time detection, identification, and localization of FODs, are essential tasks to ensure the safety of air transportation. Currently, the methods widely used for detecting FOD include manual inspection [2], radar detection technology [3–5], and optical detection technology [6–8]. The manual inspection method may be somewhat effective to some extent, but it lacks efficiency and reliability. This method is unable to keep up with the growing demand for frequent takeoffs and landings on airport runways, and it also poses a risk of secondary invasion of foreign objects. Radar detection technologies such as millimeter wave radar and LiDAR have high detection accuracy and precise localization, which are conducive to detecting FOD. However, these products have exorbitant manufacturing and maintenance costs, which limit their practical application. Additionally, some challenges exist, such as complex signal processing procedures and inadequate information regarding target characteristics. Traditional optical image processing techniques rely on the color and geometric features of the target and use methods such as image difference, wavelet transform, edge feature

extraction, and morphology to process and extract the feature information of targets, achieving recognition and classification of targets. However, traditional object detection methods require much time in feature design, and manually designed features have low robustness to target diversity issues. Moreover, traditional image processing techniques are susceptible to variability in imaging conditions such as lighting, occlusion, and defects. As a result, significant variations in the output of these techniques can be observed. This adversely impacts the precision and accuracy achievable in target recognition applications.

FOD detection is a challenging task of finding small targets in complex backgrounds [9]. Airport road surfaces are complex and varied, with oil stains, pavement textures, and objects obscuring them. Many researchers from different countries have proposed various detection algorithms based on optical imaging technology. With the advancement of computer hardware, deep learning algorithms, particularly convolutional neural networks (CNN), have experienced significant progress. The detection accuracy of traditional methods is much lower than that of deep learning methods, so most of the current research in target detection is focused on convolutional neural networks. It is also necessary to study intelligent FOD detection algorithms based on images, which can overcome the limitations of manual inspection, radar detection technology, and traditional optical detection technology. Girshick et al. [10] proposed the classic two-stage object detection algorithm R-CNN (Region Convolutional Neural Network); Redmon et al. [11] proposed the one-stage object detection algorithm You Only Look Once (YOLO); and Liu et al. [12] proposed a single shot multibox detector (SSD) combining the anchor mechanism of Faster R-CNN [13] and YOLO's regression idea. The aforementioned methodology utilizes visible light band imagery of the target as the input data source. Based on extracted visual features, this enables global detection and recognition of the image. Visible light images have rich color and texture features, so most deep learning detection algorithms can achieve good detection accuracy for medium to large targets with sparse distribution in visible light images.

When the target is tested with a small size or in low lighting conditions, such as during the night, visible light cannot effectively capture the image of the target. Many domestic and international researchers have proposed methods for the fusion and recognition of infrared and visible light images to address the challenge of detecting small targets in visible light images [14–17]. Compared with existing radar detection systems, infrared imaging technology is less susceptible to interference from electronic devices and can compensate for the limitations of visible light detection systems in smoke, fog, and night vision scenarios. Currently, fusion strategies for dual light data can be categorized into data-level fusion, decision-level fusion, and feature-level fusion. The main idea of data-level fusion is to decompose infrared and visible images with different filters and to fuse feature vectors containing common characteristics at the pixel level or regionally [18,19]. Image fusion at the data level necessitates stringent requirements regarding the resolution, pixel alignment, image quality, positional displacement, and additional aspects of the constituent images. Failure to satisfy these prerequisites results in substantial fusion errors. Figure 1c demonstrates image fusion using PCNN (Pulse Coupled Neural Network) [20] for feature extraction followed by fusion, while Figure 1d shows fusion based on SIFT (Scale Invariant Feature Transform) [21] feature point extraction. While these techniques can effectively fuse and denoise infrared and visible images, they degrade image quality for scenes containing small targets. Specifically, the fused result fails to enhance the feature information of the small targets and actually reduces discernible information about them. The fused image not only does not enhance the feature information of small targets but also reduces the information of small targets. The target detection method by Bai et al. [22] uses convolutional neural networks to detect targets in infrared and visible light images, respectively, and combines the detection results through weighted fusion to improve detection performance. However, this method only fuses the detection results and lacks complementary feature information. Ning et al. [23] also proposed a decision-level fusion algorithm for target detection of visible and infrared images. Although a model-based reliable fusion strategy was used instead of the weighted fusion strategy proposed by Bai,

the detection effect was improved. However, the essence was still to fuse the detection results of the two models, lacking the extraction and complementarity of target feature information. Zhang et al. [24] used fully convolutional neural networks to extract features from infrared and visible light images, which were then weighted. They fused the features by calculating weights under different modalities, enriching the target feature information. However, this weighting method enriches the features and introduces more noise.

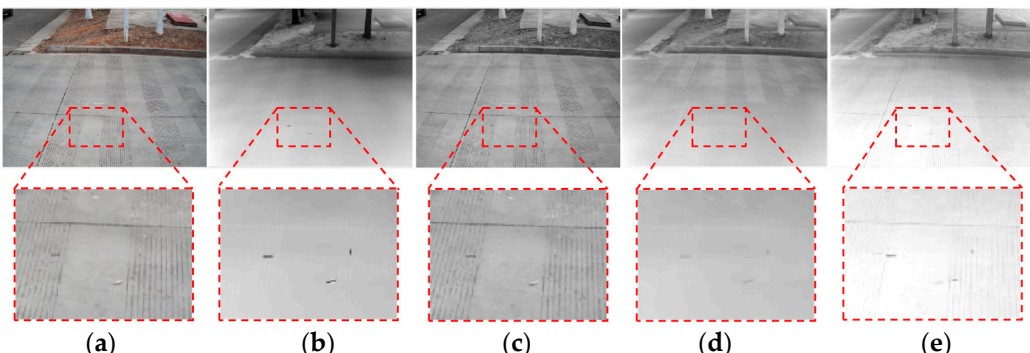

|     |     |     |     |     |
| --- | --- | --- | --- | --- |
| (**a**) | (**b**) | (**c**) | (**d**) | (**e**) |

**Figure 1.** Comparison of results of different fusion methods (The image inside the red box is a partially enlarged drawing): (**a**) original visible light image; (**b**) original infrared image; (**c**) PCNN fusion; (**d**) SIFT fusion; (**e**) pixel weighted average.

This paper aims to review existing research results and effectively utilize target information in infrared and visible light images to improve the accuracy and robustness of small-scale FOD detection tasks in complex environments. A dual-mode small-scale FOD detection algorithm, which integrates multiple attention mechanisms, was proposed for this purpose. By improving the input layer of the model, the original infrared image and visible light image are input into the model at the same time so that the network can extract more and more original infrared and visible light feature information. Reduced noise generation and enriched feature extraction information compared to previous studies. And a large number of comparative experiments were conducted on collected datasets to demonstrate the superiority of the proposed method.

The remaining sections of this paper are structured as follows: Chapter 2 introduces the intelligent dual light camera and YOLOv5 model in detail, as well as this paper's proposed improvement method. Chapter 3 presents the experiments, including the dataset construction and model performance comparisons. Finally, Chapter 4 summarizes the paper.

## 2. Methods

### 2.1. Dual Light Mode

This paper uses an intelligent dual light camera to collect images of related foreign objects on the airport runway road surface. The image acquisition equipment consists of a vanadium oxide uncooled infrared focal plane detector and a 1/2.7-inch two-megapixel high-performance CMOS visible light detector. The detailed parameters are shown in Table 1.

**Table 1.** Intelligent dual light camera equipment parameters.

| Parameter | Infrared Sensor | Parameter | Visible Sensor |
| --- | --- | --- | --- |
| Focal Length | 15 mm | Focal Length | 6 mm |
| 17 μm field of view angle | 40.5° × 33.0° | Resolution Ratio | 1920 × 1080 |
| 12 μm field of view angle | 28.6° × 23.3° | White Balance | Support |
| Monitoring range | 25 m–75 m | Day Night Conversion | Automatic switching |

### 2.2. Benchmark Network

The single-stage object detection network YOLOv5 proposed by Glenn Jocher [25] in 2020 combines the advantages of the previous YOLO series algorithms, resulting in more robust detection accuracy and speed. This paper selects YOLOv5, which has a faster speed and smaller network model, as the benchmark network. The model structure and detailed structure of each component are shown in Figure 2.

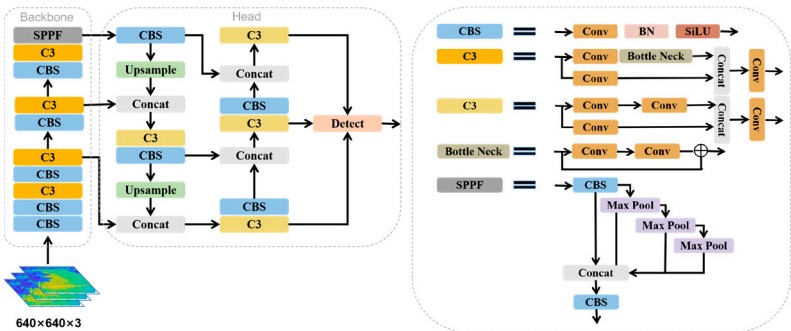

**Figure 2.** Structural diagram of benchmark network.

The backbone of YOLOv5 uses CSPNet (Cross Stage Partial Network), which enhances the learning ability of the network, maintaining accuracy while being lightweight enough [26]. The backbone network mainly comprises the CBS module, C3 module, and SPPF (SPP-Fast) module. The CBS module includes a Convolutional Layer (Conv), a Batch Normalization Layer (BN), and a SiLU activation function to extract information features from images at different levels. The C3 module includes three Conv and multiple residual components, optimizing the path of gradient backpropagation and improving network learning ability while reducing computational costs and memory overhead. The SPPF module draws inspiration from spatial pyramid pooling (SPP) [27], expands the receptive field, and extracts the essential contextual features with a faster detection speed. The prediction head is a module that generates features of different scales on images of different scales, enabling the detection of multi-scale targets. It comprises two components: a Feature Pyramid Network (FPN) [28] and a Path Aggregation Network (PAN) [29]. FPN is a feature extractor that outputs proportionally sized feature maps at multiple levels, while PAN is a feature enhancer that boosts information flow between different levels of the feature pyramid. Due to the combination of semantic information conveyed by FPN from top to bottom and positional information transferred by PAN from bottom to top, the YOLOv5 network has strong feature fusion ability.

### 2.3. Improvement Method

#### 2.3.1. Overall Framework

Given the small scale of FOD targets, the utilization of visible light imagery for detection introduces difficulties in discriminating the target from background clutter. For example, it is difficult to distinguish between metallic (nuts, screws) foreign objects and cement concrete backgrounds with similar color characteristics. Particularly in nighttime or dim conditions, the efficacy of visible light imaging is severely compromised. Numerous established and widely utilized object detection algorithms are not directly transferrable for the task of identifying foreign objects on airport runways. Therefore, we introduce infrared images to compensate for the shortcomings of visible light images. However, the fusion between infrared and visible light images can be challenging and requires careful design to effectively enhance the feature information of targets. On the contrary, such approaches may introduce interference that obscures small targets, precluding the network from sufficiently focusing on subtle features. The proposed methodology is illustrated in Figure 3, which outlines the overall framework. Firstly, image fusion of infrared and visible modalities is not implemented during preliminary data processing. Instead, both

infrared and visible images are input simultaneously in the network's input layer. This enables the preservation of more low-level small target information at the input level. Then, we introduce our multiple attention module (represented by MAM in the diagram), which is designed for small targets in the benchmark network and uses multiple attention mechanisms to enhance the extraction of small target feature information. Finally, the bounding box regression loss function is designed as CIOU loss to improve the positioning accuracy of small targets.

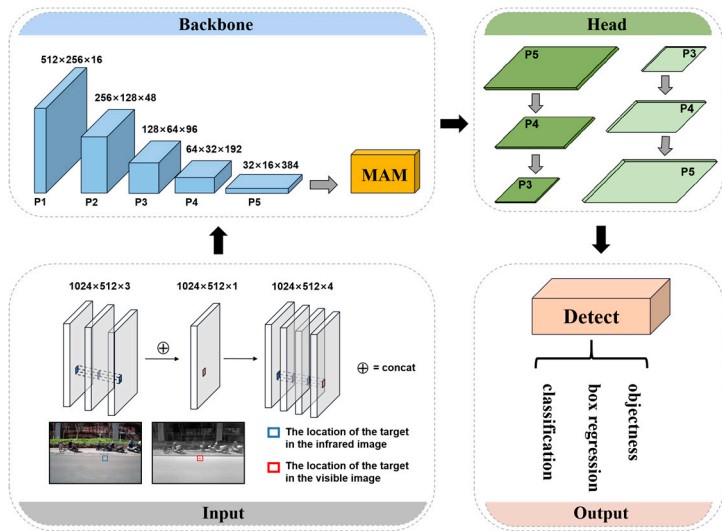

**Figure 3.** Network structure.

### 2.3.2. Attention Module

The attention mechanism is widely used in computer vision tasks, with its efficacy in enhancing model performance well established. Currently, the mainstream attention mechanisms can be divided into channel, spatial, and self-attention. The purpose of channel attention is to capture the relationship between different channels (feature maps). The weight of each feature channel is automatically obtained through network learning. Then, different weight coefficients are assigned to each channel to reinforce important features and suppress non-important ones. Channel attention mechanisms, such as SE (Squeeze and Excitation) [30], have been shown to be effective in lightweight network research. Spatial attention is intended to enhance the expression of features in critical areas. Its essence is to transform the spatial information in the original picture to another space and retain the critical information through the spatial conversion module. It generates a weight mask for each position and outputs it with weighting, enhancing the specific target area of interest while weakening irrelevant background areas. Attention mechanisms can help models focus on important features better, improving model performance. The CBAM (Convolutional Block Attention Module) proposed by Woo et al. [31] combines attention mechanisms from two dimensions: feature channel and feature space.

This paper proposes a multi-attention module to enhance the feature extraction of small-scale targets, reduce background interference, and overcome the lack of visible light images in dim environments. This module was added to the backbone to improve the detection performance of the model in complex environments. Figure 4 shows the structure of the module. It consists of two channel attention modules and one spatial attention module. The features extracted from the previous layer are fed into the first channel attention module, which outputs a channel-wise weighted feature map. The spatial attention module then takes this feature map as input and outputs a spatially weighted feature map. The second channel attention module further refines the feature map by applying another channel-wise weighting. The final output is the refined feature map. Figure 5 illustrates the detailed structure of each sub-module.

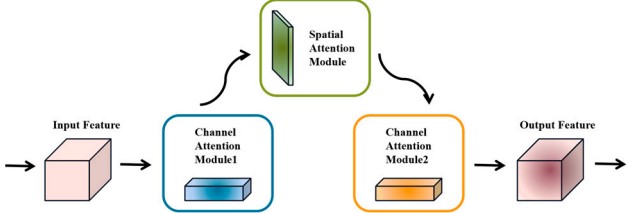

**Figure 4.** Structure diagram of multiple attention module.

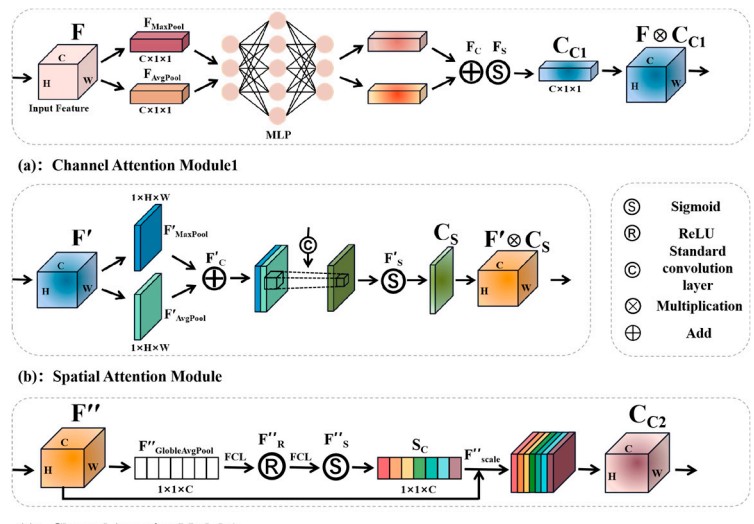

**(a): Channel Attention Module1**

**(b): Spatial Attention Module**

**(c): Channel Attention Module2**

**Figure 5.** Detailed structure diagram of attention module.

The loss of information is first reduced by using parallel pooling, which consists of a maximum pooling and an average pooling to represent the maximum pooled features and average pooled features, respectively. Maximum pooling is a technique that selects the maximum value within a local region of a feature map. It can preserve the texture features, which are the most important information for image recognition. The maximum pooling unit is sensitive to the local maximum value, which means it can enhance the contrast and reduce the noise in the feature map. The input mixed feature map $F \in R^{C \times H \times W}$ includes visible and infrared feature maps, and different descriptors, $F_{MaxPool} \in R^{C \times 1 \times 1}$ and $F_{AvgPool} \in R^{C \times 1 \times 1}$, are generated using maximum pooling and average pooling. Then, these two descriptors are fed into a multi-layer perceptron (MLP) for learning, concatenating the output results of the MLP. Then, the sigmoid function is used for mapping to obtain the channel attention map $C_{C1} \in R^{C \times 1 \times 1}$, $C_{C1}$ and $F$ is input to obtain the output feature map $F' \in R^{C \times H \times W}$ through replication. The calculation of the channel attention module is shown in Equation (1).

$$C_{C1}(F) = Sig(MLP(MaxPool(F)) + MLP(AvgPool(F)))$$
$$F' = C_{C1} \otimes F \tag{1}$$

The location information of the target cannot be ignored, so we use a spatial attention module right after the first channel attention module. Firstly, the output of the first channel attention module is taken as the input of the spatial attention module. The maximum pooling $F'_{MaxPool} \in R^{1 \times H \times W}$ and average pooling $F'_{AcgPool} \in R^{1 \times H \times W}$ operations are applied along the channel axis, cascade feature descriptors, and standard convolution $f^{7 \times 7}$ is used to generate spatial attention mapping $C_S$. The calculation equation for the spatial attention module is shown in Equation (2).

$$C_S(F') = Sig(f^{7 \times 7}(concat([MaxPool(F'); AcgPool(F')])))$$
$$F'' = C_S \otimes F' \tag{2}$$

In addition, since visible light images have little impact in dim environments, infrared images contain more information. In other words, in dim environments, the weight of infrared images should be as high as possible to be more conducive to object detection. So, after the spatial attention module, we added a second channel attention mechanism to enhance feature representation capabilities further. Firstly, the feature map output by the spatial attention module is reduced to a scalar $S_C \in R^{1 \times 1 \times C}$ in the channel dimension through a global average pooling operation. Then, the scalar is mapped into a channel attention vector through two Fully Connected Layers (FCL), which are used to weigh the channels of the feature map and obtain the attention feature map $C_{C2} \in R^{W \times H \times C}$.

$$F''_{GlobleAvgPool} = \frac{1}{W \times H} \sum_{i=1}^{W} \sum_{j=1}^{H} F''(i,j) \tag{3}$$

$$F''_{Scale}(F'', S_C) = F'' \cdot S_C \tag{4}$$

*2.4. Evaluating Indicator*

In deep learning object detection, to verify the superiority and effectiveness of algorithms, detection accuracy, inference speed, model size, and frames per second (frame rate) are usually selected as the leading indicators for evaluating model performance. Therefore, this paper adopts the following commonly used evaluation indicators in object detection: (1) precision (P); (2) recall (R); (3) average precision (AP) of a single category; (4) mean average precision (MAP); (5) frames per second (FPS) is used to measure the model comprehensively.

$$Precision = \frac{TP}{TP + FP} \tag{5}$$

$$Recall = \frac{TP}{TP + FN} \tag{6}$$

Among them, $TP$ (True Positive) is the number of targets correctly detected by the model, $FP$ (False Positive) is the number of targets for model error detection, $FN$ (False Negative) is the correct number of targets missed by the model.

The mean average precision (MAP) for a set of queries is the mean of the average precision scores for each query, as shown in Equation (7).

$$MAP = \frac{1}{c} \sum_{i=1}^{c} AP_i = \frac{1}{c} \sum_{i=1}^{c} \int_0^1 P(R) dR \tag{7}$$

$$AP = \int_0^1 P(R) dR \tag{8}$$

Among them, $AP$ represents the accuracy of each category, and $c$ represents the number of categories in the dataset.

The number of frames detected per second refers to the average value of the model's detection speed for foreign object images in the test set, as shown in Equation (9).

$$FPS = \frac{F_T}{T_C} \tag{9}$$

Among them, $F_T$ represents the total number of frames, and $T_C$ represents the total time for model detection.

*2.5. Boundary Box Regression Loss Function*

The dataset established in this paper contains mainly small-scale FOD images, where the pixel area is very small compared to the whole image. The default GIoU localization loss function in the YOLOv5 network degenerates into IoU when two boxes have an inclusion relationship, revealing the drawback of the IoU loss function, which is that small distance

movements after the inclusion relationship do not reduce loss, leading to convergence difficulties. To improve the model's accuracy in locating foreign objects on airport runways, the default GIoU localization loss function inherent to the YOLOv5 network architecture is supplanted with a CIoU loss function. The distance, overlap rate, scale, and penalty terms between the target and the frame anchor are taken into account by CIoU, which can make the target box regression more stable. A schematic diagram of the Complete CIoU is illustrated in Figure 6. The blue solid box denotes the predicted bounding box, the orange solid box denotes the ground truth bounding box, the white dotted box represents the smallest enclosing bounding box, with c indicating the diagonal distance of the minimum enclosing box. The CIoU equation is defined in (14).

$$IoU = \left( \frac{b \cap b^{gt}}{b \cup b^{gt}} \right) \tag{10}$$

$$Loss_{GIoU} = 1 - \left( IoU - \frac{c + b \cup b^{gt}}{|c|} \right) \tag{11}$$

$$v = \frac{4}{\pi^2} \left( arctan \frac{w}{h} - arctan \frac{w^{gt}}{h^{gt}} \right)^2 \tag{12}$$

$$\alpha = \frac{v}{(1 - IoU) + v} \tag{13}$$

$$Loss_{CIoU} = 1 - \left( IoU - \frac{\rho^2 \left( b, b^{gt} \right)}{c^2} - \alpha v \right) \tag{14}$$

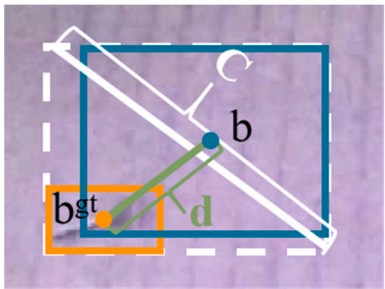

**Figure 6.** CIoU loss schematic.

Among them, the prediction box's midpoint and the target box's midpoint are represented by $b$ and $b^{gt}$, while $\rho^2 \left( b, b^{gt} \right)$ represents the distance between them. Based on the positional relationship between the two boxes, the box that exactly covers them is defined as the minimum bounding box, with a diagonal length of $c.v$ reflects the similarity in aspect ratio between the two boxes. $w$ and $w^{gt}$ represent the width of the prediction box and target box, respectively. $h$ and $h^{gt}$ represent the height of the prediction box and target box, respectively. $\alpha$ represents the weight function, which is the influencing factor of $v$. Applying the CIoU loss function can reduce the target positioning error in the small target intrusion detection task of airport runway foreign objects and enhance the fitting ability of the prediction box to the genuine target box.

## 3. Experiment

In this section, we describe the experimental design process and provide an analysis of the experimental results. Firstly, we introduce the experimental setup and configuration. Then, a detailed description and analysis are conducted to establish the experimental dataset, and the evaluation indicators are introduced. Finally, a comparison is made between the attention mechanism proposed and various other attention mechanisms. Additionally, a visual comparison is made to demonstrate its superior performance in terms of both detection accuracy and speed.

### 3.1. Experimental Configuration

All experiments were conducted based on the Pytorch1.12.1, with hardware consisting of an NVIDIA RTX-3060Ti GPU and a 4.9 GHz (Inter Core i7-12700F) CPU. The program was run on the Windows 11 operating system, with Python version 3.8.16, and accelerated model training using CUDA11.1 and cuDNN8.0.4 during the experiment.

Throughout the experiments, we unified the input image size to 640 × 640 and added batch size set to 2. In the training phase, the number of iterations is set to 200 epochs, and the initial learning rate is set to 0.01. In the first three epochs, the learning rate of each iteration is updated using warm-up learning rate preheating to improve the convergence speed of model loss. After the three epochs, the learning rate is attenuated using the cosine annealing method to ensure the stability of model loss convergence, and the weight attenuation rate is set to 0.0005. Finally, after a comprehensive comparison of the accuracy and inference speed of the training model, the optimal model is selected.

### 3.2. Experimental Dataset Creation and Analysis

FOD detection presents a challenge when dealing with small targets set against complex backgrounds. These small-scale targets, owing to their diminutive physical dimensions or the considerable distance from the imaging source, occupy a relatively minuscule portion of the image, often encompassing only a few dozen pixels or even fewer. In the context of the COCO dataset [32], small targets are defined as those encompassing dimensions less than 32 × 32 pixels. Additionally, according to a definition from the International Society for Optical Engineering [33], a small target constitutes any target encompassing less than 80 pixels within a 256 × 256-pixel image. In simpler terms, if the size of the target is less than 0.12% of the original image area, it qualifies as a small target. The visual representation of a small target is shown in Figure 7.

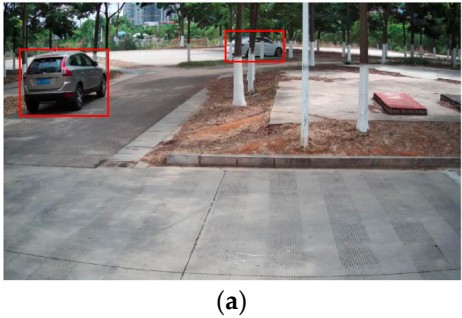 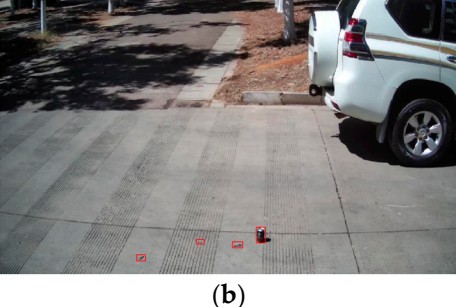

| (**a**) | (**b**) |

**Figure 7.** Instances of small target: (**a**) conventional size target (cars); (**b**) small target (screw, key, nut and bottle).

FOD refers to any foreign objects, debris, or substances that have the potential to cause damage to the aircraft. These can include metal components [34], crushed stones, paper products, animals, plants, and more. Among them, metal components are ingested by the engine, and it is very easy to cause accidents. Based on the distribution of foreign objects in actual airport runway scenes and the technical characteristics of deep learning for sample requirements, an Infrared-Visible Foreign Object Debris Dataset (IVFOD) was designed and constructed. This dataset contains four types of foreign objects: screw; nut; key; bottle.

The IVFOD collected in this experiment used cement concrete pavement and asphalt pavement on the Chenggong campus of Kunming University of Science and Technology to simulate the airport runway pavement. The dual light camera was installed on the equipment, and a twisted pair cable was utilized to interface the camera to the computer. The designated shooting angle was 30°, set on the roadside and on the road. However, data collection was conducted over multiple sessions spanning several days; thus, there may have been a deviation of ±1°–2° in the actual shooting angle for each session. Foreign objects at a distance of 5 m (±1–10 cm), 10 m (±1–10 cm), and 15 m (±1–10 cm) from the

equipment were photographed in the morning, afternoon, and evening. The entire data collection system is shown in Figure 8.

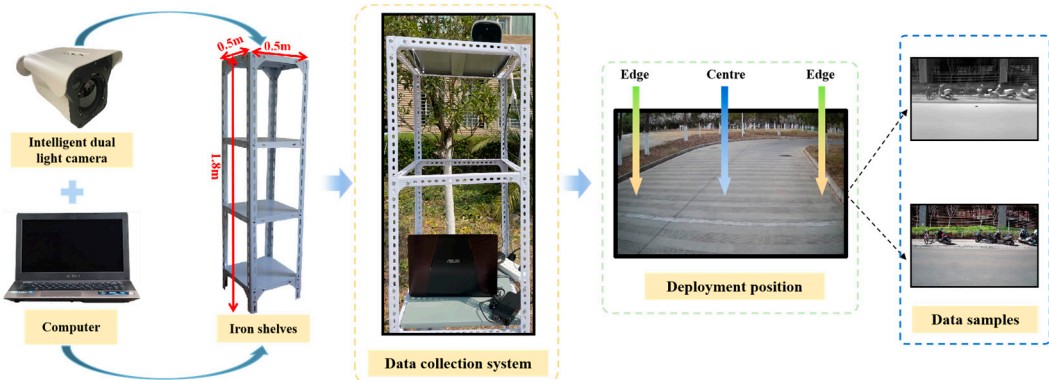

**Figure 8.** Data collection process.

Firstly, screened the target images captured by the dual light camera and finally determined the number of images in the dataset to be 4137 pairs. Then, 7217 instance targets containing four classes were labeled using Labelimg and converted into PASCAL VOC2007 annotation format files. Using the Random library, 4137 pairs of images were randomly divided into a training set and a validation set in an 8:2 ratio for model training and validation, respectively. In addition, there were 100 sets of images to verify the generalization ability of the final model. The original images and annotation images of FOD are shown in Figure 9.

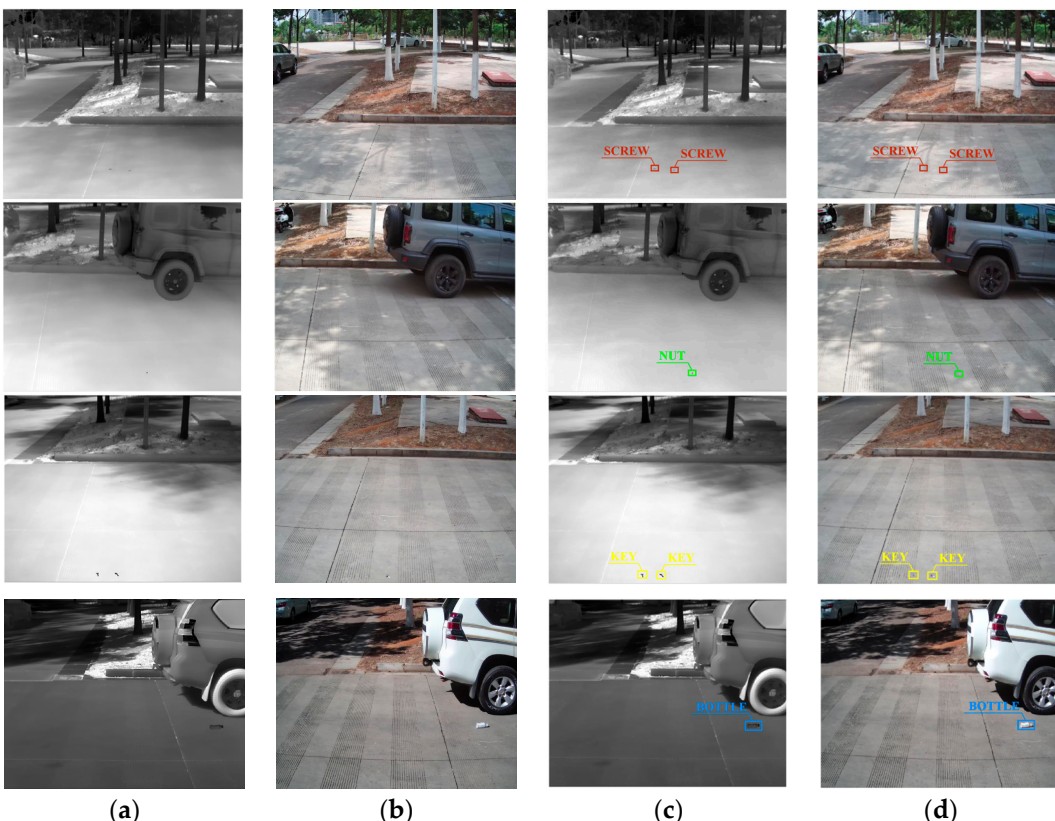

**Figure 9.** IVFOD dataset images and annotations: (**a**) original infrared image; (**b**) original visible light image; (**c**) infrared annotation map; (**d**) visible annotation map.

The size and distribution of targets used in this experiment are shown in Figure 10. The pixel area encompassed by the targets acquired in this experimental investigation uniformly

comprises less than 5000 pixels, and the size of the target is less than 0.12% of the original image area, thereby satisfying the aforementioned criteria for small target classification. In addition, the numbers of the four types of targets collected in this experiment were 2093, 1822, 1768, and 1534, respectively, with a relatively balanced distribution.

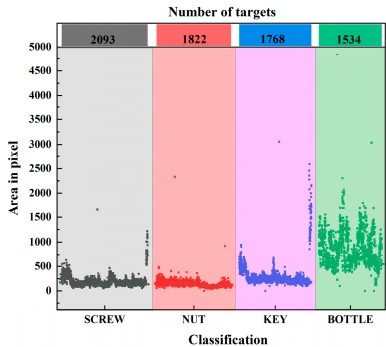

**Figure 10.** Data size and distribution.

### 3.3. Experimental Dataset Creation and Analysis

This paper compares the MAP of several different attention mechanisms under the YOLOv5 framework, including SA (Spatial Attention), CA (Channel Attention), CBAM, and SE. PURE indicates a situation where no attention mechanism has been introduced. According to Table 2, the introduction of a spatial attention mechanism reduced the model's accuracy by 5.8%. The model that used only channel attention achieved an accuracy that was 6.1% higher than the model that used mixed attention, CBAM, indicating the importance of channel attention for detecting small targets.

**Table 2.** Comparison test results of attention mechanism.

|  | Precision | Recall | MAP | Mean Loss |
|---|---|---|---|---|
| PURE | 0.671 | 0.719 | 0.713 | 0.004258 |
| SA | 0.613 | 0.784 | 0.71 | 0.004475 |
| CA | 0.722 | 0.754 | 0.747 | 0.004151 |
| CBAM | 0.661 | 0.802 | 0.763 | 0.005930 |
| SE | 0.745 | 0.838 | 0.817 | 0.004216 |
| Proposed | 0.845 | 0.913 | 0.911 | 0.004054 |

With the above experimental results, it can be seen that on the IVFOD dataset, the method proposed in this paper can effectively improve the recognition accuracy of the model for small targets. Compared with the original YOLOv5 model, the accuracy and generalization ability are both improved to some extent. The training convergence speed and accuracy of the network are also improved after introducing the CIoU loss function. A comparison of the loss functions CIoU and GIoU is shown in Figure 11c. The improvements proposed in this paper are in line with the expected results. In order to show a more intuitive comparison of the actual detection performance of each attention mechanism, Figure 11a,b give the comparison plots of the detection accuracy and loss curves of several models.

Table 3 compares the mean average precision of different objects under different models. In the case of visible images alone (single mode), the effect of increasing the depth of the network on improving the accuracy of small target detection is not very significant. In contrast, for objects such as bottles, which are significantly larger than the other three types of objects, deepening the depth of the network can improve detection accuracy. By comparing the first and third sets of data in the table, it can be observed that the addition of infrared images improved the detection accuracy for small targets, even without increasing the network depth. Moreover, the detection accuracy for tiny targets such as nuts was enhanced by using infrared images. SLBAF-Net [35] has added an infrared and visible light adaptive fusion layer to the front end of the network, which enhances feature extraction

while fusing infrared and visible image feature information, effectively improving the accuracy of small target detection. In order to better preserve the target information in the original image, the model in this paper chooses to directly input the original infrared and visible light images in the initial stage of the network. Subsequently, this paper's newly established attention module was added to the backbone to improve the accuracy of small target detection further. The experimental results show that the model in this paper has achieved good results in detecting four types of targets: bottle, screw, key, and nut.

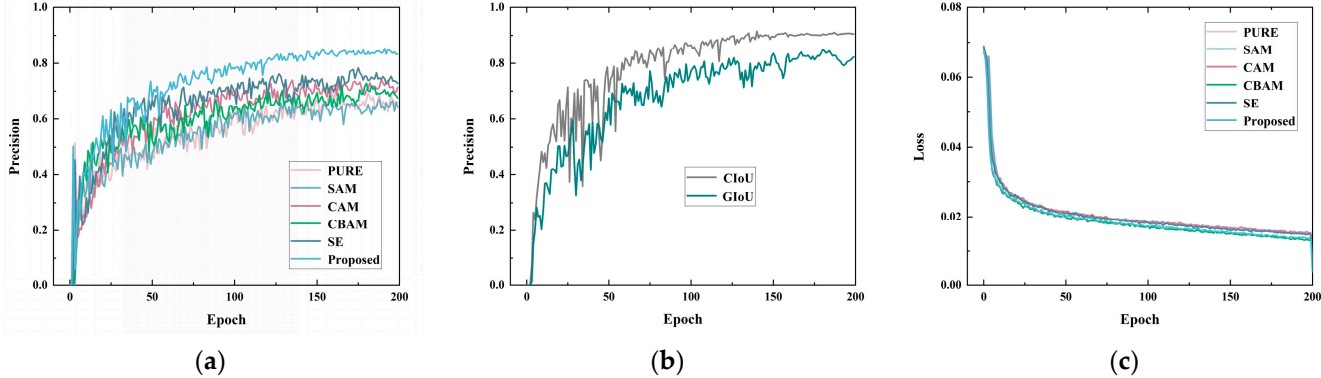

**Figure 11.** Comparison of attention precision and loss curve and loss function: (**a**) the variation curve of precision of different attention mechanisms with training epochs; (**b**) loss curves of different attention; (**c**) the variation curve of precision of different loss functions with training epochs.

**Table 3.** MAP values for different categories and models.

| Model | Input | Bottle | Screw | Key | Nut |
|---|---|---|---|---|---|
| YOLOv5s | Vis | 0.934 | 0.617 | 0.837 | 0.478 |
| YOLOv5m | Vis | 0.944 | 0.623 | 0.861 | 0.484 |
| YOLOv5s | Inf and Vis | 0.942 | 0.608 | 0.665 | 0.634 |
| YOLOv5m | Inf and Vis | 0.966 | 0.819 | 0.854 | 0.708 |
| SLBAF-Net | Inf and Vis | 0.938 | 0.843 | 0.851 | 0.768 |
| Proposed | Inf and Vis | 0.958 | 0.905 | 0.953 | 0.827 |

The specific training results for each model are shown in Table 4. To furnish a comprehensive evaluation of detection performance, operational efficiency, and resource demands, MAP, frames per second (FPS), and parameters were employed as quantification metrics.

**Table 4.** Test results of different models.

| Model | Input | MAP | FPS (frame/s) | Parameters | Training Time | Discrimination Time |
|---|---|---|---|---|---|---|
| YOLOv5s | Vis | 0.717 | 36.9 | 12.5 M | 10.524 h | 3.8 ms |
| YOLOv5m | Vis | 0.728 | 27.66 | 47.0 M | 10.911 h | 6.9 ms |
| YOLOv5s | Inf and Vis | 0.713 | 35.5 | 14.4 M | 9.958 h | 3.3 ms |
| YOLOv5m | Inf and Vis | 0.837 | 38.2 | 47.3 M | 10.261 h | 5.0 ms |
| SLBAF-Net | Inf and Vis | 0.850 | 25.13 | 5.7 M | 10.730 h | 16.1 ms |
| Proposed | Inf and Vis | 0.911 | 36.3 | 42.2 M | 10.246 h | 4.4 ms |

When comparing the experimental results of training YOLOv5s and YOLOv5m on the visible dataset for detecting small foreign objects, simply increasing the network depth does not significantly improve small target detection accuracy, with only a 1.1% gain in MAP. When comparing the experimental results of training YOLOv5s and YOLOv5m on infrared and visible light datasets, enriching the feature information of small foreign objects using combined infrared and visible images enables the deeper network to improve detection accuracy. Specifically, there is a 12.4% increase in mean average precision (MAP), indicating enhanced detection across various foreign object types overall. SLBAF-Net has an infrared

and visible light adaptive fusion layer added to the front end of the network, which enhances feature extraction and fuses the infrared and visible image features, effectively improving small target detection accuracy. The model in this paper achieves a MAP of 91.1%, which is better than other models and has the highest detection accuracy for the target. FPS reached 36.3, with a slightly slower detection speed than the baseline model. While achieving high accuracy, the parameters of the improved model are 42.2 M, which proves that our model is lightweight enough, easy to deploy, and can be applied to real-time detection scenarios.

The original infrared image is shown in Figure 12a, and the original visible light image is shown in Figure 12b. Small target detection based on visible light images suffers from the challenges of tiny pixels, low resolution, unclear target texture features, and ineffective color features due to the small size of the target. In poor lighting conditions, such as at night, visible light loses its edge completely.

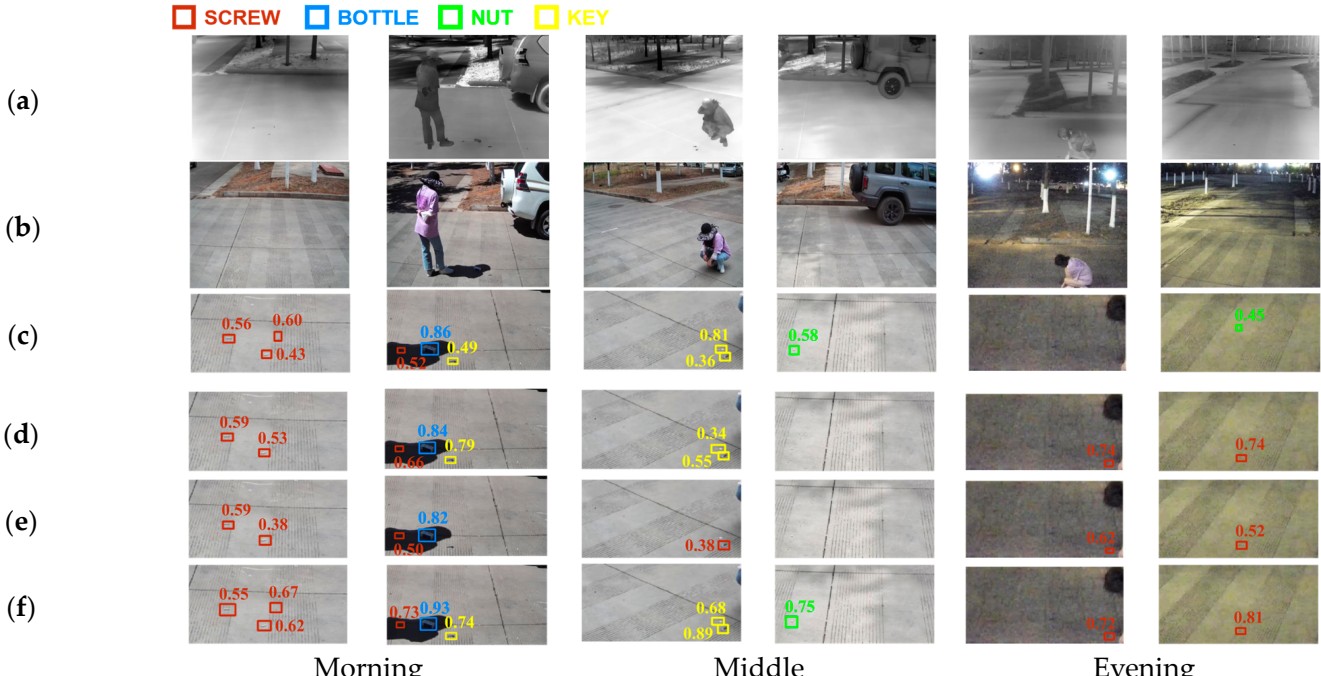

**Figure 12.** Sample diagram of detection results: (**a**) original infrared image; (**b**) original visible light image; (**c**) YOLOv5m—input visible light image; (**d**) SLBAF-Net—input visible light image and infrared image; (**e**) YOLOv5m—input visible light image and infrared image; (**f**) proposed—input visible light image and infrared image.

The six exemplar image pairs were acquired at three distinct temporal instances—morning, afternoon, and evening. Some results from different networks are shown in Figure 12c–f. Given the diminutive scale of the small targets, the resultant detections were cropped to showcase only the localized areas encompassing the targets.

Visible light images convey more information about the target under favorable lighting conditions. However, there are still many things that could be improved by relying on visible images for small target detection. Figure 12c shows the case of using only visible light images as input to the network. Visible light images possess non-negligible advantages, but the fifth and sixth images show missed and wrong detection. So, visible light images have non-negligible drawbacks in poor lighting conditions, such as at night. Figure 12d shows the use of SLBAF-Net with both visible and infrared images as inputs to the network. SLBAF-Net effectively improves the network's performance in detecting small targets at night. However, as shown in the first figure, the target detection accuracy is unsatisfactory. The third result plot in Figure 12d shows that the confidence level of correctly detecting the target as a key is 0.34 and 0.55, respectively, which is a decrease of 0.47 and an increase of

0.19, respectively, when compared to the third result plot in Figure 12c. Figure 12e shows the use of YOLOv5m with both visible and infrared images as inputs to the network. The first and fourth images both show missed detection. The third image shows the phenomenon of wrong detection. Moreover, the detection accuracy is less satisfactory than SLBAF-Net. Figure 12f shows the result images using the method proposed. Based on reducing wrong and missed detection, it effectively improves the accuracy of network detection of small targets. The third result plot in Figure 12f shows that the confidence level of correctly detecting the target as a key is 0.68 and 0.89, respectively, which are increased by 0.34 compared to the third result plot in Figure 12d. Further, it improves the detection accuracy of small targets at night.

## 4. Conclusions

The existing deep learning algorithms still have some problems when applied to FOD detection tasks. In visible light images, the targets are minuscule, with texture and color features that are not prominent, and the background features are noisy and diverse. This paper addresses the above problems and draws the following conclusions:

1.  In order to make up for the problem of missing target information in visible images, the method of complementing the information of visible images and infrared images is adopted to enrich the feature information of small targets;
2.  A multi-attention mechanism is proposed to suppress the interference of background features for effective recognition of small targets. This method not only improves the accuracy of small target detection during the day but also takes into account the detection performance of small targets at night. The model detection speed can also meet the real-time detection demand of FOD;
3.  A comprehensive experimental validation shows that the method in this paper is superior to other methods for detecting small-scale FOD. Although the model parameters are not the lightest, they are still at a high level and meet expectations in terms of quantitative results and visualization effects. The proposed model achieves a detection speed of 36.3 frame/s. The model also attains an average accuracy of 91.1%, which is 7.4% higher than the baseline network.

The method in this paper has a promising application in practical FOD detection. In the future, we plan to augment the datasets across more weather variability (rain, snow, hail, etc.), broader foreign object scales and categories, as well as expanded lighting parameters (intensities, source types, spectra). Considering global airport diversity, targeted studies will also be initiated with high-altitude sites like Kunming Changshui Airport. These efforts seek to continuously boost model generalization and adaptability to novel environments.

**Author Contributions:** Conceptualization, H.X.; methodology, Y.M.; software, Y.M.; validation, C.C. and P.L.; investigation, C.C. and W.H.; resources, W.H.; writing—original draft preparation, Y.M.; writing—review and editing, H.X. and L.W.; visualization, Y.M.; supervision, L.W.; project administration, H.X.; funding acquisition, H.X. All authors have read and agreed to the published version of the manuscript.

**Funding:** This work was supported by the National Natural Science Foundation of China (No. 12262015).

**Institutional Review Board Statement:** Not applicable.

**Informed Consent Statement:** Not applicable.

**Data Availability Statement:** The data presented in this study are available on request from the corresponding author. The data are not publicly available due to privacy.

**Conflicts of Interest:** The authors declare no conflicts of interest.

## Abbreviations

| | |
|---|---|
| FOD | Foreign Object Debris |
| IVFOD | Infrared-Visible Foreign Object Debris Dataset |
| CIoU | Complete Intersection over Union |
| GIoU | Generalized Intersection over Union |
| MPA | Mean Pixel Accuracy |
| CNN | Convolutional Neural Network |
| SiLU | Sigmoid Linear Unit |
| MLP | Multi-Layer Perception |
| FPS | Frames Per Second |

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
