# Peer review of "Small-Scale Foreign Object Debris Detection Using Deep Learning and Dual Light Modes"

_applsci, doi:10.3390/app14052162_

Round 1
Reviewer 1 Report
Comments and Suggestions for Authors
This is an interesting manuscript on a deep learning-based intelligent detection approach for addressing the problem of low accuracy and efficiency in small-scale FOD detection. A review of previous research results, as well as the efficient use of target information in infrared and visible light images to increase the accuracy and robustness of small-scale FOD detection tasks in complicated situations, is also presented. This interesting topic is within the scope of the Applied Sciences Journal.
Some concerns regarding your manuscript need to be resolved as follows:
a) Consider the title's wording
b) Substitute the word formula for equation.
c) The parameters in equations 10–13 should be specified more precisely.
d) Ensure that the figures 8 and 9, accurately represent the intended information. Please improve details.
e) Please, provide a clear and more quantitative description of the remaining probable causes of error in the suggested method.
f) It is necessary to include a quantitative comparison of the performance of different approaches' planning.
Comments on the Quality of English LanguageModerate editing of English language required
Reviewer 2 Report
Comments and Suggestions for Authors
The authors investigates the opportunity to implement Artificial Intelligence solutions among the airport's infrastructure for runway surface.
The paper fits the Applied Sciences profile on perfect way.
Some aspects to review:
* The introduction provides sufficient background and includes enough references. However, the part dedicated to the literature review is not clear (being found under Introduction chapter), which make it hard to understand the relationship between this work and the previous research.
* Line 122: the reference to Glenn Jocher (2020) research is missing.
* The limitations of the proposed model can be better highlighted.
However, the part dedicated to research contributions (methods and experiments) is very well outlined and concretely explains how a smart dual-light camera and Deep learning techniques can be integrated into the entire process of automatic airport runway surface scanning.
In conclusion, the paper is well constructed and can be published after minor revision.
Reviewer 3 Report
Comments and Suggestions for Authors
The authors present a paper entitled "Small-scale FOD Detection Based on Deep Learning and Dual Light Mode". In brief, they use deep learning alongside dual light mode imaging for enhancing the detection of small-scale FOD. Their approach is interesting, aiming to address a critical safety issue in air transportation. However, in order to potentially further enhance the paper's contribution to its field, the following comments for major-revision are suggested:
1. Validation Against Diverse Environmental Conditions: While the paper provides a comprehensive approach to FOD detection using a dual light camera system and an improved YOLOv5 model, it would benefit from a broader validation across diverse environmental conditions. Testing the model in various weather conditions, lighting (beyond just day and night), and runway surface types could strengthen the argument for the model's robustness and practical applicability.
2. Comparison with State-of-the-Art Methods: The paper presents an improvement over the baseline YOLOv5 network; however, a more detailed comparison with other state-of-the-art FOD detection methods could offer a clearer perspective on the proposed method's advantages and limitations. This includes both traditional methods and other deep learning approaches, highlighting the unique contributions of the dual light mode and multi-attention mechanism.
3. Scalability and Deployment Considerations: The paper mentions the model's detection speed and accuracy, which are crucial for real-time detection. However, discussions on scalability, integration with existing airport surveillance systems, and any potential deployment challenges would be valuable. This includes hardware requirements, the process for updating the model with new data, and handling false positives.
4. Future Work on Model Generalization: While the paper successfully demonstrates the model's effectiveness on the IVFOD dataset, future research directions could focus on generalizing the model to detect a broader range of FOD types and sizes. Additionally, exploring the potential for the model to adapt to new environments without extensive retraining would be of interest, considering the global diversity of airports
Comments on the Quality of English LanguageThere are no comments for English language
Round 2
Reviewer 1 Report
Comments and Suggestions for Authors
I have no further comments
Reviewer 3 Report
Comments and Suggestions for Authors
The authors have satisfactorily addressed the concerns raised, providing comprehensive experimental validation and comparisons with state-of-the-art methods. Their response demonstrates significant improvements in their model's robustness and practical applicability. Moreover, their future work promises to further enhance model scalability and generalization. Based on these considerations, I recommend accepting the paper for publication